# Enzyme-Enhanced Codelivery of Doxorubicin and Bcl-2 Inhibitor by Electrospun Nanofibers for Synergistic Inhibition of Prostate Cancer Recurrence

**DOI:** 10.3390/ph15101244

**Published:** 2022-10-10

**Authors:** Zheng Liu, Xing Luo, Yongxin Mo, Pengkai Zhao, Haixia Wang, Youqiang Fang, Yanteng Xu

**Affiliations:** Laboratory of Biomaterials and Translational Medicine, Center for Nanomedicine, Department of Urology, The Third Affiliated Hospital, Sun Yat-Sen University, Guangzhou 510630, China

**Keywords:** electrospun fiber, DOX, Bcl-2, ABT199, prostate cancer, tumor recurrence

## Abstract

One of the great challenges of postoperative prostate cancer management is tumor recurrence. Although postoperative chemotherapy presents benefits to inhibit unexpected recurrence, it is still limited due to the drug resistance or intolerable complications of some patients. Electrospun nanofiber, as a promising drug carrier, demonstrating sustained drug release behavior, can be implanted into the tumor resection site during surgery and is conductive to tumor inhibition. Herein, we fabricated electrospun nanofibers loaded with doxorubicin (DOX) and ABT199 to synergistically prevent postoperative tumor recurrence. Enzymatic degradation of the biodegradable electrospun nanofibers facilitated the release of the two drugs. The primarily released DOX from the electrospun nanofibers effectively inhibited tumor recurrence. However, the sustained release of DOX led to drug resistance of the tumor cells, yielding unsatisfactory eradication of the residual tumor. Remarkably, the combined administration of DOX and ABT199, simultaneously released from the nanofibers, not only prolonged the chemotherapy by DOX but also overcame the drug resistance via inhibiting the Bcl-2 activation and thereby enhancing the apoptosis of tumor cells by ABT199. This dual-drug-loaded implant system, combining efficient chemotherapy and anti-drug resistance, offers a prospective strategy for the potent inhibition of postoperative tumor recurrence.

## 1. Introduction

One of the greatest challenges in life science is how to overcome cancer [1,2,3]. Prostate cancer is one of the most prevalent malignancies in the male population and there are approximately 360,000 deaths because of prostate cancer each year. So far, the predominantly preferred treatment in clinical practice for prostate cancer is surgical resection [1]. However, there still exist small residual tumors after surgery, which may lead to tumor recurrence and even metastasis [4]. It is reported that the biochemical recurrence rate of prostate cancer after radical resection is as high as 53% [5,6]. Conventional adjunct chemotherapy after surgery, which remains a general strategy for cancer treatment, has played a critical role in eliminating residual tumors up to now [7]. However, after metabolism via blood and tissues, drugs actually acting on tumors are rare because of their low selectivity [8]. Excessive amounts of administration are often required to ensure that sufficient drugs are delivered to the tumor site [9]. At the same time, long-term chemotherapy also leads to serious suffering of patients and unexpected side effects [10]. Exploring new drug vehicles, improving delivery efficacy, and ultimately eliminating residual tumors are still desperately demanded.

Localized drug delivery systems (LDDSs), which can be implanted into the tumor resection site upon surgery for precise drug delivery and sustained release, have presented promising potential in the treatment of solid tumors [11,12,13,14,15]. LDDSs offer the superior chemotherapeutic effect of the loaded drugs via minimizing the drug diffusion into other tissues compared to intravenous or other systemic administrations. As of now, various types of LDDSs, such as hydrogel, microneedle, exosome and so on, have been created and explored [16,17,18,19]. Electrospun fibers, a kind of superfine nanofiber with a thickness from the nanometer-to-micrometer range, exhibit excellent characteristics because of its high drug-loaded ability, sustained drug release, great stability, reliable biosafety and convenient preparation [20]. These advantages make electrospun nanofibers an ideal LDDS which has been widely studied in cancer therapy, wound healing, cell culture and other biomedical applications [21]. DOX is a classical chemotherapy drug with a broad spectrum of antitumor activity. The mechanism of antitumor capability is mainly due to inhibiting the duplication of nucleic acid in tumor cells via embedding double-stranded DNA molecules [22]. Recently, DOX-loaded electrospun fibers were reported to have efficient drug delivery [23,24,25]. Even though DOX-loaded nanofibers have shown remarkable tumor inhibition, there are still some deficiencies. On the plus side, the localized drug can accurately target tumor cells, but at the same time, tumor cells might survive with accumulated mutations under prolonged chemotherapy. Therefore, the tumor may become resistant to chemotherapy and start growing again, accompanied by a reduced efficacy of chemotherapeutic drugs [26]. As a consequence, it is an urgent requirement to investigate and address the conundrum of DOX resistance.

The B-cell Lymphoma 2 (Bcl-2) protein family plays an important role in regulating the balance of pro-apoptotic and anti-apoptotic regulatory factors [27]. The overexpression of Bcl-2 occurs in a variety of tumor cells, and Bcl-2 participates in the maintenance of tumor growth [28,29]. Meanwhile, high Bcl-2 expression has been reported to inhibit the sensitivity of tumor cells to therapeutic drugs [30]. It has been found that Bcl-2 overexpression significantly hampered DOX-induced apoptosis [31]. To address this drug resistance issue, Bcl-2 protein inhibitors are being studied and manufactured for oncology therapy, which have obtained promising achievements in patients with virous cancers [32]. ABT199 (Venetoclax) is a kind of potent and highly selective Bcl-2 inhibitor, which shows strong affinity (Ki < 0.01 nM) for Bcl-2 proteins but with very weak affinity for other apoptotic proteins [33]. Many clinical trials have begun using ABT199 as a single- or multi-drug treatment for various tumors (breast cancer, lung cancer, prostate cancer, lymphoma and so on), which shows great prospects in increasing tumor sensitivity to chemotherapeutic drugs [32,34,35]. Therefore, the Bcl-2 inhibitors can be considered as beneficial adjuvants to overcome the drug resistance of different tumors.

Herein, we developed an effective and secure combination strategy by using DOX and an ABT199-loaded poly(lactic-co-glycolic acid) (PLGA) electrospun nanofiber system for localized drug delivery to prevent postoperative tumor recurrence (Figure 1). The combined administration of dual drugs demonstrated excellent synergistic anti-tumor effects in vitro and in vivo. By implanting the DOX-loaded nanofibers after tumor resection, the sustainedly released DOX was able to inhibit tumor recurrence, however, which inevitably induced drug resistance via promoting Bcl-2 overexpression. To overcome the drawback of drug resistance, the co-administration of ABT199, a Bcl-2 inhibitor, was simultaneously released from the nanofibers, which deactivated the Bcl-2 protein and thereby enhanced the apoptosis of drug-resistant tumor cells. Therefore, ABT199 maintained the high sensitivity of tumor cells against DOX, yielding an efficient eradication of the residual tumors. Our designed dual-drug-loaded electrospun nanofiber system posits a new strategy for the exploitation of safe and effective postoperative tumor therapy.

## 2. Results and Discussion

### 2.1. Characterization of Electrospun Nanofibers

Poly (lactic-co-glycolic acid) (PLGA) was used to manufacture electrospun fibers because of its biocompatibility and excellent fiber-forming properties [36]. The color of blank PLGA electrospun fiber membrane (Fibers) was white, as shown in Figure 2a. When DOX was loaded (DOX@Fibers), the fibers appeared orange. The ABT199-loaded fibers (ABT199@Fibers) were light yellow, and the appearance of DOX-ABT199-loaded fibers (DOX-ABT199@Fibers) were between DOX@Fibers and ABT199@Fibers. Scanning electron microscopy (SEM) was used to observe the nanoscale morphology of the electrospun nanofibers. It can be seen from Figure 2b that the nanofibers were smooth and had a uniform structure without entanglement, which indicated that the fibers were quite stable because the uniform shape ensured a steady release of the loaded drug [37]. The diameter of blank fibers was about 570 nm, and it did not change obviously after loading DOX. However, when the fibers were loaded with ABT199, the thickness became even thinner. This may be due to the change of electrical conductivity of the polymer solution after adding ABT199 [20,38]. It is noted that the diameter of the nanofibers was calculated by measuring more than 100 fibers in the SEM images (Figure 2c). Collectively, the diameter of Fibers, DOX@Fibers, ABT199@Fibers, and DOX-ABT199@Fibers was about 570, 520, 310 and 360 nm, respectively.

### 2.2. The Release of DOX and ABT199 from Electrospun Nanofibers In Vitro

The drug release curve (Figure 3a) shows that 24% of DOX was released within 100 h in PBS at pH 6.5. It should be noted that a PBS buffer with pH 6.5 at 37 °C was selected as the release buffer to simulate the acidic environment of a tumor [39]. Conversely, 18% of ABT199 (Figure 3b) had been released within 100 h, a relatively slower release rate than DOX, which may be due to their different water solubility [40]. This slow and persistent release of both drugs offered the possibility of topical drug delivery after surgery, alleviating the pain and side effects caused by frequent administration. The tumor environment is complicated, not only fairly acidic, but also rich in lipase and other metabolic substances [39]. We added lipase (20 U/mL) into the release system to better mimic the release behavior of drugs from the electrospun nanofibers. With the addition of lipase, the accumulative release of DOX and ABT199 approached 51% and 35% within 100 h with the help of the enzymatic degradation of the electrospun nanofibers, respectively. Furthermore, the accumulative release of DOX and ABT199 within 24 h reached 25% and 21%, respectively. This means that abundant drugs were released within the first 24 h, which was conducive to killing most of the residual tumors left after the surgery in the early stage of drug-loaded material embedding [41]. The durable drug release indicated that our designed dual-drug-loaded nanofibers might be an utmost-potential platform to excellently inhibit tumor recurrence.

### 2.3. Synergistic Effect of DOX and ABT199 In Vitro

We next explored the synergistic antitumor effect of combining DOX with ABT199 via evaluating the viability of the two drugs treating RM-1 cells by using CCK-8 kit assay. To determine the optimum dosage ratio of DOX and ABT199, the dosage of DOX was firstly fixed, and the ratio of them was set as 1:0, 1:0.1, 1:1, and 1:10. Figure 4a shows that after 12 h incubation of RM-1 cells with DOX@Fibers containing 5 µg/mL DOX, i.e., the dose ratio of DOX and ABT199 was 1:0, about 53% of cells had survived. When the concentration of ABT199 was less than 5 μg/mL (0, 0.5 and 5 µg/mL), the cell viability upon treatment with ABT199 and DOX was not obviously decreased compared to treatment with only DOX@Fibers. However, when the concentration of ABT199 increased to 50 µg/mL (ratio of 1:10), the cell viability significantly decreased from 53% to 26%. Meanwhile, we checked the activity of RM-1 cells treated with ABT199 alone for 12 h. As shown in Figure 4b, ABT199 alone showed a negligible cytotoxic effect on RM-1 cells up to 50 µg/mL. In order to assess the synergistic effect of DOX and ABT199, the combination index (CI) was calculated [42]. When the dosage ratio of DOX and ABT199 was 1:10, the CI reached 1.7, demonstrating the highest synergistic effect among the different ratios we investigated (Figure 4c). After determining the dosage ratio of DOX and ABT199, immunofluorescence staining was performed to observe the expression level of Bcl-2 in RM-1 cells with different treatments after 24 h. Figure 4d shows that Bcl-2 was overexpressed in RM-1 cells treated with DOX@Fibers, and the expression of Bcl-2 in RM-1 cells was negligible after ABT199@Fibers treatment. Furthermore, when RM-1 cells were treated with DOX combined with ABT199 (DOX-ABT199@Fibers), Bcl-2 was still in a low observed level. This finding suggested that DOX induced Bcl-2 overexpression, yielding the possibility of drug resistance during chemotherapy. Fortunately, ABT199 could effectively overcome drug resistance and amplify the tumor-suppressive effect of DOX. As a consequence, with the help of ABT199, DOX-ABT199@Fibers presented a synergistic effect, which enhanced the chemotherapeutic effect of DOX.

### 2.4. Antitumor Effect of Drug Loaded Electrospun Nanofibers In Vitro

To further evaluate whether the combination of DOX and ABT199 could effectively improve the cytotoxicity, Live/Death staining assay was applied to observe the viability of RM-1 after different treatments (Control, Fibers, DOX@Fibers, ABT199@Fibers, DOX-ABT199@Fibers). Calcein AM is a fluorescent dye for staining live cells, which has membrane permeability [25]. After entering cells, it will be hydrolyzed by intracellular esterase and bind with intracellular calcium ions, producing strong green fluorescence. Because dead cells lack esterase, calcein AM can only label live cells. PI is generally used to screen apoptotic cells [25]. It cannot pass through the membrane of live cells; instead, it can cross through the membrane of dead cells and embed in double-stranded DNA, accompanied with emitting red fluorescence. As shown in Figure 5a, the RM-1 cells in the Control, Fibers and ABT199@Fibers groups looked healthy (stained with Calcein AM in green) without dead cells (stained with PI in Red). About half of RM-1 cells were killed in the DOX@Fibers group, while even more cells were dead in the DOX-ABT199@Fibers group. When the administration time was extended to 24 h, some RM-1 cells still survived after DOX treatment. Remarkably, live tumor cells were rarely seen after DOX-ABT199@Fibers treatment. In addition, we also quantified the cell viability in each group via CCK-8 kit assay (Figure 5b), which was consistent with the Live/Dead staining. It was worth noting that after 24 h treatment with blank fiber, the viability of RM-1 cells slightly increased compared to the control group, which may be due to the stimulation of cell proliferation by lactic acid of PLGA [43].

### 2.5. Antitumor Recurrence In Vivo

To verify the effect of dual-drug-loaded electrospun fibers on inhibiting postoperative tumor recurrence in vivo, a tumor recurrence model was firstly constructed on C57 mice. Briefly, the tumor model was constructed by subcutaneous injection of 5 × 10^6^ RM-1 cells into each mouse. After 10 days, the tumor was surgically removed with 8 mm^3^ left to simulate postoperative recurrence. Different materials (PBS as the control, DOX+ABT199, Fibers, DOX@Fibers, ABT199@Fibers, and DOX-ABT199@Fibers) were immediately implanted into the incision upon tumor resection, and the tumor volume was monitored continuously (Figure 6a). Figure 6c,d show that the combination of free DOX with free ABT199 can only slightly inhibit tumor recurrence. This may be due to the fast diffusion of free drugs to the surrounding tissues, resulting in rapid metabolism and unsatisfactory antitumor effect [44]. Compared to the PBS group (G1), blank fibers (G3) had little effect on antitumor recurrence. Notably, both DOX-loaded (G4) and ABT199-loaded (G5) fibers showed better tumor inhibition compared to the free drug group (G2). However, single-drug-loaded fibers could not efficiently inhibit tumor recurrence, as all the tumors were recurrent in both G4 and G5 (Figure 6b). Remarkably, dual-drug-loaded fibers (G6) demonstrated excellent postoperative tumor suppression, as tumors in 50% of mice were completely eradicated. The tumor inhibition process could be described as follows. DOX was continuously released from the electrospun fiber to inhibit tumor recurrence. Nevertheless, the prolonged effect of DOX increased the resistance of tumors, resulting in the overexpression of Bcl-2 in tumor cells. Therefore, those tumor cells with drug resistance survived even against continuously released DOX. Fortunately, ABT199 released from the fibers could specifically bind to Bcl-2 in tumor cells to trigger apoptosis, amplifying the chemotherapeutic effect and thus attenuating drug resistance of DOX. By using PLGA-nanofiber as a carrier, DOX and ABT199 efficiently inhibited postoperative tumor recurrence in an ingenious synergistic manner. In order to visualize the tumor size, the recurrent tumors of all groups were dissected out and captured, as shown in Figure 6b. Due to the potential errors in the volume measurement, tumors in each group were weighed for further comparison. As shown in Figure 6e, the tumor weight was 5.99 ± 4.15 g in the PBS group vs. 0.10 ± 0.13 g in the DOX-ABT199@Fibers group. The significant difference between the above two groups confirmed the successful tumor inhibition by our designed DOX-ABT199@Fibers. In addition, compared to the PBS group, the tumor suppression rates of ABT199@Fibers, DOX@Fibers and DOX-ABT199@Fibers were 48 ± 12%, 77 ± 17% and 98 ± 2%, respectively (Figure 6f). Therefore, DOX-ABT199@Fibers, with a great tumor suppressive effect, may be considered as a promising candidate for clinical postoperative therapy.

## 3. Materials and Methods

### 3.1. Materials

ABT199 was obtained from Acon Biotech Co., Ltd. (Hangzhou, China). DOX was obtained from Bide Medical Technology Co., Ltd. (Shanghai, China). Poly (lactic-co-glycolic acid) (PLGA) was purchased from Boli Biomaterials Company (Shenzhen, China). Lipase was obtained from Macklin (Shanghai, China). Hexafluoroisopropyl alcohol (HFIP) was purchased from Aladdin (Shanghai, China). The Bcl-2 antibody was purchased from Affinity Biosciences (Liyang, China). CoraLite594–conjugated Goat Anti-Rabbit IgG was purchased from Proteintech (Wuhan, China). FlouroshieldTM with DAPI was obtained from Sigma-Aldrich (St. Louis, MO, USA). CCK-8 kit was purchased from APExBIO Technology LLC (Houston, TX, USA). The male C57 mice were purchased from Sijiajingda Biotechnology Co., Ltd. (Guangzhou, China).

### 3.2. Fabrication of the Electrospun Nanofibers

To manufacture the DOX-loaded nanofibers, DOX was firstly dissolved in HFIP. Then, PLGA (PLGA/HFIP = 3:25, *w*/*v*) was added to the drug-loaded system and stirred overnight to obtain a homogeneous solution. Next, the DOX-PLGA solution fulfilled in a syringe was slowly injected through a needle, and a voltage of 10–12 kV was applied between the needle and the spinning receiver. The flow rate was set as 0.96 mL/h and the distance between needle and receiver was about 10 cm. After adjusting the above parameters, DOX-loaded nanofibers were fabricated upon the evaporation of HFIP. The manufacturing procedure of blank nanofibers and ABT199-loaded nanofibers were basically the same as the preparation of DOX-loaded nanofibers. To obtain different ratios of DOX and ABT199 in the nanofibers, the solution of DOX and ABT199 in HFIP were separately prepared. Then, we mixed them in predetermined proportions (DOX: ABT199 = 1:0.1, 1:1, 1:10, *w*/*w*). After mixing thoroughly, DOX-ABT199-loaded fibers were prepared using the same method.

### 3.3. Characterization of the Electrospun Nanofibers

Scanning electron microscopy (SEM, Oxford Quanta 400F, Oxford, UK) was used to observe the morphology and structure of nanofibers. Before observation, the fiber surface was sprayed with gold three times to increase its conductivity. After adjusting the air pressure of the sample chamber to 3 × 10^−4^ mBar, the surface structure of the scaffold was observed. ImageJ software was used to measure the diameter of the nanofibers.

### 3.4. The Release of DOX and ABT199 from the Electrospun Nanofibers In Vitro

The release of DOX was detected by a multifunctional microplate reader (Synergy H1MF, BioTek, Winooski, VT, USA). First, 2 mg drug-loaded electrospun nanofibers were immersed in 2 mL PBS (pH 6.5) at 37 °C. Subsequently, the solution was withdrawn and monitored to evaluate DOX release, which was quantified by detecting the fluorescence intensity at 595 nm under excitation wavelength of 500 nm. Meanwhile, the release of ABT199 was evaluated by a UV-Vis spectrophotometer (UV-2700, Shimadzu, Kyoto, Japan), which was quantified by detecting the absorbance value at 207 nm. The release of DOX and ABT199 was further evaluated in PBS (pH 6.5) with lipase (20 U/mL).

### 3.5. Synergistic Antitumor Effect of DOX and ABT199 In Vitro

A total of 3 × 10^4^ RM-1 cells per well in a 48-well plate were treated with ABT199-loaded fibers containing ABT199 at different concentrations (0, 0.5, 5 and 50 µg/mL). At the same time, fixing the concentration of DOX (5 µg/mL), different dosages of DOX and ABT199 (1:0, 1:0.1, 1:1, 1:10) were used to treat RM-1 cells for 12 h. Next, the CCK-8 kit agent was added to the culture medium after dilution with a factor of 20. After incubation for 2 h at 37 °C, the absorbance value of each well at 450 nm was detected by a microplate reader. The absorbance values were used to evaluate the activity of RM-1 cells in each group. To study the synergistic antitumor effect of DOX and ABT199, the combination index (*CI*) was calculated by using the following equation:$$CI=\frac{X_{DA}}{X_{D}+X_{A}}$$
where *X_DA_* is the antitumor efficiency of DOX combined with ABT199, *X_D_* is the antitumor efficiency of DOX, and *X_A_* is the antitumor efficiency of ABT199.

### 3.6. Immunofluorescence Assay of the Bcl-2 Expression

Immunofluorescence assay was applied to explore the effect of different administrations on the expression of Bcl-2 in RM-1 cells. First, 8 × 10^4^ RM-1 cells cultured on each coverslip in a 24-well plate were treated with PBS, DOX (5 µg/mL), ABT199 (50 µg/mL), and DOX + ABT199 (5 µg/mL DOX, 50 µg/mL ABT199), respectively. After incubation for 24 h, the cultured cells were fixed with 3% paraformaldehyde for 20 min, followed by permeabilization with 0.25% triton for 20 min. Then, 10% goat serum, Bcl-2 antibody, CoraLite594–conjugated Goat Anti-Rabbit IgG, and flouroshield^TM^ with DAPI were successively used to treat the above cells. Afterwards, a fluorescence microscope (Ti2-U, Nikon, Tokyo, Japan) was used to evaluate the expression of Bcl-2 in the RM-1 cells by observation of the fluorescence of CoraLite594.

### 3.7. Antitumor Effect In Vitro

The antitumor effect was evaluated via CCK-8 assay and Live/Dead cell staining. A total of 3 × 10^4^ RM-1 cells per well in a 48-well plate were treated with PBS, blank fibers, DOX-loaded fibers (5 µg/mL), ABT199-loaded fibers (50 µg/mL), and DOX-ABT199-loaded fibers (5 µg/mL DOX, 50 µg/mL ABT199), respectively. After 12 h, Live/Dead cell staining was performed with calcein AM and propidium iodide (PI), which was excited at 470 nm and 555 nm by the illumination of the microscope, respectively. The obtained fluorescence images were used to estimate the survival state of RM-1 cells. Meanwhile, CCK-8 kit assay was utilized to statistically evaluate the cell viability of RM-1 cells. The same method was used to assess the activity of RM-1 cells after 24 h.

### 3.8. Antitumor Recurrence In Vivo

The animal procedures were approved by the Animal Ethics Committee of Sun Yat-sen University. All procedures complied with the Guidelines for Care and Use of Laboratory Animals of Sun Yat-sen University. Male C57 mice, 6–8 weeks, were used to construct a prostate cancer model. In short, 5 × 10^6^ RM-1 cells were subcutaneously injected into the right flank of each mouse. Ten days later, when the tumor volume reached 200–300 mm^3^, the tumor was surgically removed. It was worth noting that the tumor was resected to leave a residual volume of about 8 mm^3^ to simulate postoperative tumor recurrence. After surgical resection, various materials (PBS, blank fibers, DOX-loaded fibers, ABT199-loaded fibers, and DOX-ABT199-loaded fibers) were implanted into the tumor resection site, respectively. Subsequently, the recurrent tumors were monitored continuously for 12 days. The length (a) and width (b) of the tumor were measured using vernier calipers, and the tumor volume was calculated according to the following formula, Volume = (a × b^2^)/2. The recurrent tumor was dissected and photographed after the mice sacrifice.

### 3.9. Statistical Analysis

The statistical significance was calculated according to Student’s *t*-test. * stands for *p* < 0.05, ** stands for *p* < 0.01, *** stands for *p* < 0.001.

## 4. Conclusions

In summary, we have successfully developed and characterized DOX- and ABT199-loaded PLGA electrospun nanofibers to efficiently prevent postoperative prostate cancer recurrence. These nanofibers, with a diameter of about 360 nm, were capable of sustainedly releasing the dual drugs simultaneously. The released DOX presented inefficient inhibition of the tumor progression, owing to the upregulation of Bcl-2 expression induced decreased apoptosis of tumor cells. Remarkably, the co-delivered ABT199 can specifically bind to the Bcl-2 proteins to inhibit the activation of Bcl-2 and facilitate the tumor cell apoptosis in vitro and in vivo. With the help of ABT199, the drug resistance of residual tumor cells against DOX was significantly alleviated. Our results suggested that DOX- and ABT199-co-loaded electrospun nanofibers presented an efficiently synergistic effect to prevent tumor recurrence after prostate tumor resection, providing a promising strategy for postoperative care. Looking towards future development, the surgical treatment effect may greatly benefit from the integrated drug-loaded platform, and we will further improve our delivery platforms to conquer the drug resistance and recurrence of tumors in higher efficacy.

## Figures and Tables

**Figure 1 pharmaceuticals-15-01244-f001:**
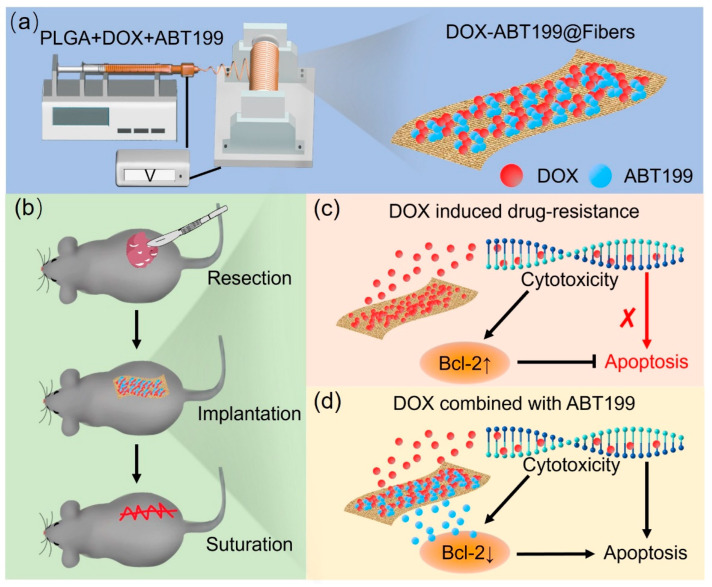
Schematic illustration of (**a**) the fabrication of DOX-ABT199@Fibers, (**b**) the procedure of the inhibition of the tumor recurrence, and (**c**,**d**) the local delivery of drugs after tumor resection and the synergistic effect of the sustainedly released DOX and the Bcl-2 inhibitor ABT199 (the red cross symbol means that the process was weakened).

**Figure 2 pharmaceuticals-15-01244-f002:**
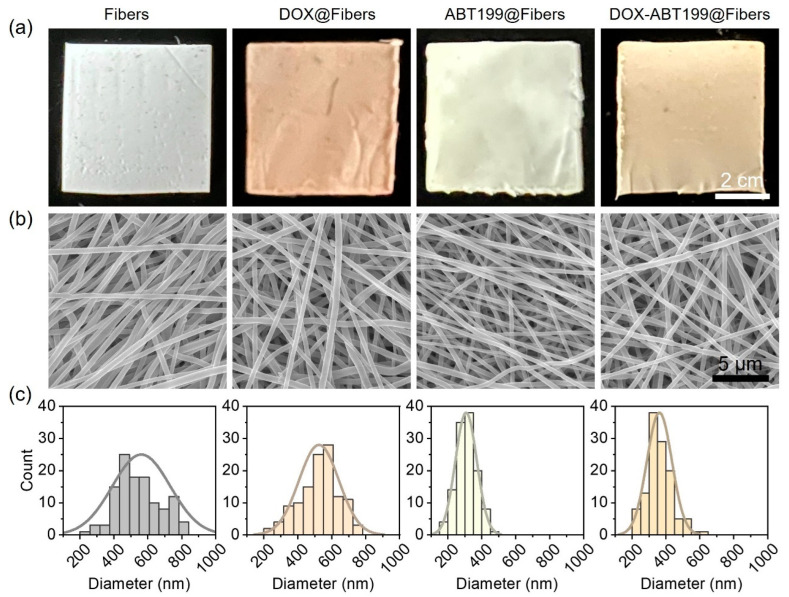
Characterization of the electrospun nanofibers loaded without and with drugs. (**a**,**b**) Digital and SEM images of Fibers, DOX@Fibers, ABT199@Fibers, and DOX-ABT199@Fibers, and (**c**) their corresponding diameter distributions extracted from the SEM images, respectively.

**Figure 3 pharmaceuticals-15-01244-f003:**
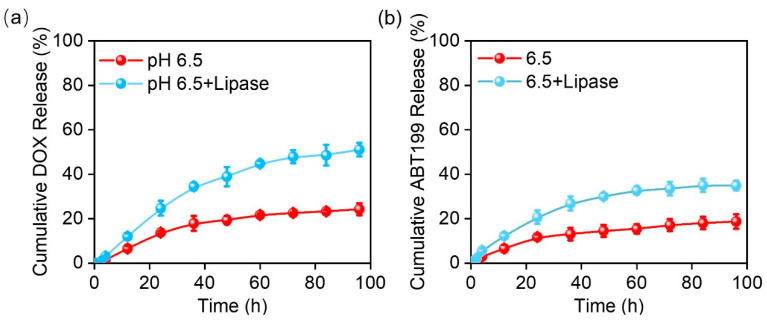
The cumulative release curves of (**a**) DOX and (**b**) ABT199 from the fabricated electrospun fibers in vitro.

**Figure 4 pharmaceuticals-15-01244-f004:**
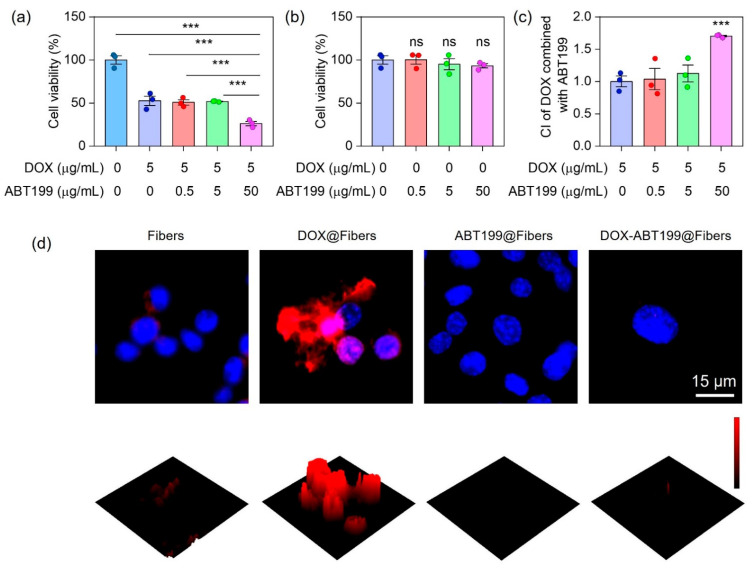
Synergistic tumor cell inhibition effect of DOX combined with ABT199 in vitro. (**a**,**b**) The cell viabilities of RM-1 cells, and (**c**) the CI of DOX combined with ABT199, calculated based on the cell viability. (**d**) The immunofluorescence images of RM-1 cells showing the Bcl-2 expression after receiving different treatments (*** stands for *p* < 0.001 and ns stands for no statistical significance).

**Figure 5 pharmaceuticals-15-01244-f005:**
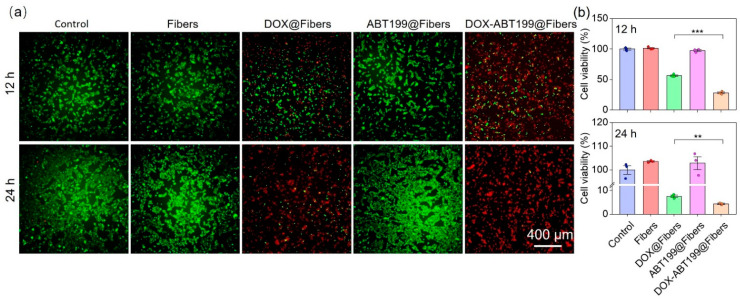
Evaluation of anti-tumor effect of electrospun fibers loaded with different components in vitro. (**a**) Live/Dead staining of RM-1 cells was performed after treatment with different materials for 12 and 24 h, and (**b**) the cell viability was evaluated via CCK-8 kit assay (*** stands for *p* < 0.001 and ** stands for *p* < 0.01).

**Figure 6 pharmaceuticals-15-01244-f006:**
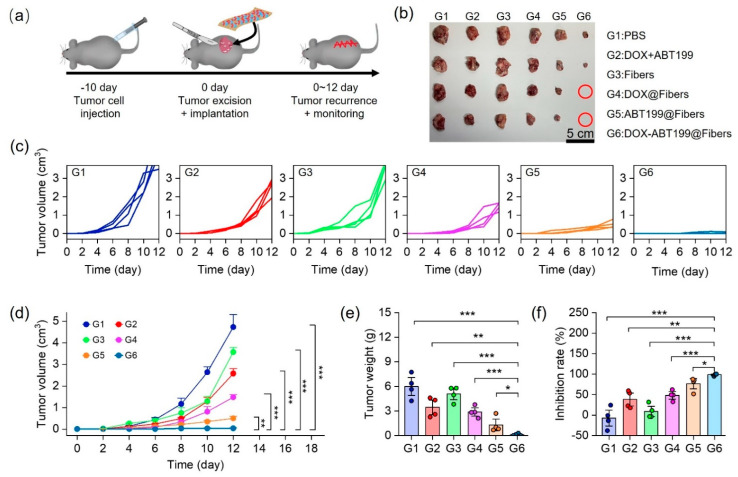
Anti−tumor recurrence in vivo. (**a**) Schematic diagram of tumor recurrence model construction, tumor resection, material implantation and the recurrent tumor monitor. (**b**) Digital images of recurrent tumors in each group (The red circle means no tumor). The individual (**c**) and summarized (**d**) volumes of recurrent tumors monitored over time after treatment in each group. (**e**) Recurrent tumor weights and (**f**) the corresponding tumor suppression rates in each group (*** stands for *p* < 0.001, ** stands for *p* < 0.01 and * stands for *p* < 0.05).

## Data Availability

Data is contained within the article.

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
