# Peer review of "Enzyme-Enhanced Codelivery of Doxorubicin and Bcl-2 Inhibitor by Electrospun Nanofibers for Synergistic Inhibition of Prostate Cancer Recurrence"

_pharmaceuticals, 2022, doi:10.3390/ph15101244_

Round 1

Reviewer 1 Report

In my opinion, the manuscript is apt for publication in this journal after the authors have addressed the following comments and modifications:

1.       Overall, the English in the paper needs to revisited. The English used in the manuscript is difficult to comprehend in  some sentences which  need to be rephrased like:   

“One of the great challenges of postoperative prostate cancer is tumor recurrence”

“All these results indicated that 1our designed dual-drug-loaded nanofibers with durable drug release behavior might be an utmost potential platform to excellently inhibit tumor recurrence.” Line 138

“However, because of the low selectivity of chemotherapeutic drugs, the amount of drugs actually acting on the tumor site is rare after metabolism via blood and tissues”.  line 38

To better reveal the drug release from the electrospun nanofibers in the

tumor environment, we added lipase (20 U/mL) into the release system. Line 131

We next explored the synergistic antitumor effect of DOX and ABT199 via evaluating 145

the cell viability of RM-1 cells by using CCK-8 kit assay. Line 145

In order to visually observing the tumor size, the recurrent tumors of all groups were dissected out. line 225

2.        Minor spelling errors like “DOX-ABT199-loaded fivers” line 326, should be rectified.

3.       It is not clear if the authors are planning to carry any further research in orthopic murine models to further validate the findings. That would be interesting to compare with currently used ectopic model.

4.       The authors have not stated the way forward and future plan post accomplishment of this research.

Author Response

Dear Reviewers,

Thank you so much for your valuable comments to our manuscript entitled “Enzyme Enhanced Codelivery of Doxorubicin and Bcl-2 Inhibitor by Electrospun Nanofibers for Synergistic Inhibition of Prostate Cancer Recurrence”. Your comments are very valuable and helpful for the improvement of our paper. According to your kind suggestions, we have revised the whole manuscript. We hope that this updated version will meet with your approval and be qualified to be published in Pharmaceuticals. All the changes to the text have been marked in red. The responses to your comments are listed as follows.

Q1: The English in the paper needs to revisited.

R1: Thank you very much for your suggestion. According to the reviewer’s suggestion, we carefully examined the grammar of the manuscript and revised the following sentences which were marked in red in the revised manuscript:

  1. The sentence ‘One of the great challenges of postoperative prostate cancer is tumor recurrence.’ was changed to ‘One of the great challenges of postoperative prostate cancer management is tumor recurrence.’ in line 12 on Page 1 of the revised manuscript.
  2. The sentence ‘All these results indicated that our designed dual-drug-loaded nanofibers with durable drug release behavior might be an utmost potential platform to excellently inhibit tumor recurrence.’ was changed to ‘The durable drug release indicated that our designed dual-drug-loaded nanofibers might be an utmost potential platform to excellently inhibit tumor recurrence.’ in line 138 on Page 4 of the revised manuscript.
  3. The sentence ‘However, because of the low selectivity of chemotherapeutic drugs, the amount of drugs actually acting on the tumor site is rare after metabolism via blood and tissues.’ was changed to ‘However, after the metabolism via blood and tissues, drugs actually acting on tumor is rare because of their low selectivity.’ in line 38 on Page 1 of the revised manuscript.
  4. The sentence ‘To better reveal the drug release from the electrospun nanofibers in the tumor environment, we added lipase (20 U/mL) into the release system.’ was changed to ‘We added lipase (20 U/mL) into the release system to better mimic the release behavior of drugs from the electrospun nanofibers.’ in line 131 on Page 4 of the revised manuscript.
  5. The sentence ‘We next explored the synergistic antitumor effect of DOX and ABT199 via evaluating the cell viability of RM-1 cells by using CCK-8 kit assay’ was changed to ‘We next explored the synergistic antitumor effect of combining DOX with ABT199 via evaluating the viability of the two drugs treated RM-1 cells by using CCK-8 kit assay.’ in line 145 on Page 5 of the revised manuscript and marked it in red.
  6. The sentence ‘In order to visually observing the tumor size, the recurrent tumors of all groups were dissected out,’ was changed to ‘In order to visualize the tumor size, the recurrent tumors of all groups were dissected out and captured,’ in line 225 on Page 8 of the revised manuscript.

Q2: Minor spelling errors like “DOX-ABT199-loaded fivers” line 326, should be rectified.

R2: Thank you very much for your helpful comment. We have revised the sentence to ‘DOX-ABT199-loaded fibers’, in line 326 on Page 10 of the revised manuscript and marked it in red.

Q3: It is not clear if the authors are planning to carry any further research in orthopic murine models to further validate the findings. That would be interesting to compare with currently used ectopic model.

R3: Thank you for your advice. We also believe that the orthoptic model will more truly reflect the tumor suppression effect of our designed system. In our future work, we will further improve our drug-loaded material and conduct further research by using orthoptic model. We also added a sentence, ‘Toward future development, the surgical treatment effect may greatly benefit from the integrated drug-loaded platform, and we will further improve our delivery platforms to conquer the drug-resistance and recurrence of tumors in higher efficacy.’, in line 347 on Page 11 of the revised manuscript and marked it in red.

Q4: The authors have not stated the way forward and future plan post accomplishment of this research.

R4: Thank you very much for your suggestion. We agree with the reviewer that it’s necessary to mention our future plans. Therefore, we added a sentence, ‘Toward future development, the surgical treatment effect may greatly benefit from the integrated drug-loaded platform, and we will further improve our delivery platforms to conquer the drug-resistance and recurrence of tumors in higher efficacy.’, in line 347 on Page 11 of the revised manuscript and marked it in red.

Reviewer 2 Report

The study deals with the analysis of effectiveness of codelivery of DOX and ABT199 from PLGA nanofibers. It is demonstrated (in vitro and in vivo) that synergetic effect of the two drugs reduces (by twice) cancer cell viability. The results are interesting and are worth to be published.

I have two questions regarding the observations reported in the paper:

1.     What is the role of enzymatic degradation of nanofibers in cumulative drug delivery? Fig. 3 demonstrates an increase in the rate of drug release due to the presence of lipase. How is this increase connected with the rate of synergetic release of both drugs and with cell viability?

2.     DOX is treated as the main anti-cancer drug, and ABT199 is employed to maintain sensitivity of cancer cells to DOX. Surprisingly, Fig. 6d shows that the tumor volume becomes lower when fibers loaded with ABT199 (without DOX) are used compared with fibers loaded with DOX. An explanation of this result is needed.

Author Response

Q1: What is the role of enzymatic degradation of nanofibers in cumulative drug delivery? Fig. 3 demonstrates an increase in the rate of drug release due to the presence of lipase. How is this increase connected with the rate of synergetic release of both drugs and with cell viability?

R1: Thank you very much for your comment.

  The enzymatic degradation will facilitate the drug release from nanofibers. PLGA is a biodegradable polymer due to the hydrolysis of its ester linkages. The loaded drugs can diffuse from the polymer fiber. At the same time, the degradation of the PLGA fibers can accelerate the release of the loaded drugs. In the bioenvironment, lipase is one of the numerous enzymes participates in the metabolism. The existence of lipase could accelerate the hydrolysis degradation of these ester bonds. In addition, the lipase content in tumor environment is abundant compared with normal tissue due to the disorder of lipid metabolism in tumor tissue. As a result, at the tumor site, drugs in PLGA fiber are more accurately released around the tumor to better kill tumor cells. The drug release behavior presents a better anti-tumor effect of the drugs and reduces the cell viability of residual tumor cells.

Q2: DOX is treated as the main anti-cancer drug, and ABT199 is employed to maintain sensitivity of cancer cells to DOX. Surprisingly, Fig. 6d shows that the tumor volume becomes lower when fibers loaded with ABT199 (without DOX) are used compared with fibers loaded with DOX. An explanation of this result is needed.

R2: Thank you very much for your comment.

Although DOX is treated as the main antitumor drug, long-term chemotherapeutic effect makes residual tumor cells resistant to the drug. It is difficult for DOX to kill tumor cells which are already resistant to chemo drugs. This resulted in a reduction of the tumor suppressive effect of DOX. Moreover, it is noted that there exists a difference between in vitro and in vivo. ABT199 can not only maintain the sensitivity of tumor cells to DOX, but also restore the programmed cell death mechanism, thereby inducing apoptosis of tumor cells [1-3]. In addition, the dosage of ABT199 (50 mg/kg) we used was 10 times than that of DOX (5 mg/kg). These are the reasons why the efficacy of ABT199 is better than that of DOX in tumor models.

[1] J.R. Whittle, F. Vaillant, E. Surgenor, A.N. Policheni, G. Giner, B.D. Capaldo, H.R. Chen, H.K. Liu, J.F. Dekkers, N. Sachs, H. Clevers, A. Fellowes, T. Green, H. Xu, S.B. Fox, M.J. Herold, G.K. Smyth, D.H.D. Gray, J.E. Visvader, G.J. Lindeman, Dual Targeting of CDK4/6 and BCL2 Pathways Augments Tumor Response in Estrogen Receptor-Positive Breast Cancer, Clin Cancer Res 26(15) (2020) 4120-4134.

[2] Y. Liang, S. Jeganathan, S. Marastoni, A. Sharp, I. Figueiredo, R. Marcellus, A. Mawson, Z. Shalev, A. Pesic, J. Sweet, H. Guo, D. Uehling, B. Gurel, A. Neeb, H.H. He, B. Montgomery, M. Koritzinsky, S. Oakes, J.S. de Bono, M. Gleave, A. Zoubeidi, B.G. Wouters, A.M. Joshua, Emergence of Enzalutamide Resistance in Prostate Cancer is Associated with BCL-2 and IKKB Dependencies, Clin Cancer Res 27(8) (2021) 2340-2351.

[3] G. Radha, S.C. Raghavan, BCL2: A promising cancer therapeutic target, Biochim Biophys Acta Rev Cancer 1868(1) (2017) 309-314.